



# Changes in Supraglacial Lakes on George VI Ice Shelf, Antarctic Peninsula: 1973-2020

Thomas J. Barnes[1], Amber A. Leeson[1,2], Malcolm McMillan[1,3,4], Vincent Verjans[1], Jeremy Carter[1], Christoph Kittel[5]

[1]Lancaster Environment Centre, Lancaster University, Lancaster, UK.
[2]Data Science Institute, Lancaster University, Lancaster, UK
[3]Centre for Polar Observation and Modelling, Lancaster University, Lancaster, UK, LA1 4YW, UK
[4]Centre for Excellence in Environmental Data Science, Lancaster University, Lancaster, UK, LA1 4YW, UK
[5]Department of Geography, UR SPHERES, University of Liège, Liège, Belgium

*Correspondence to*: Thomas J. Barnes (t.barnes5@lancaster.ac.uk)

**Abstract.** High densities of supraglacial lakes have been associated with ice shelf instability and collapse. 2020 was a record melt year on George VI ice shelf with ~12% of its northernmost portion being covered by lakes. We use 208 Sentinel-2 and Landsat-1-8 satellite images from the past 47 years, together with climate data and firn modelling, to assess the long-term presence of lakes on George VI, thus placing 2020 within a historical context. We find that the ~12% lake coverage observed in 2020 is not unprecedented and similar to previous high lake years; events of similar magnitude occurred at least five times previously. Secondly, we find lake coverage is controlled by a combination of melting, accumulation, firn air content and firn build-up strong melting alone does not entail high lake coverage. Instead, while melting contributes positively to lake formation, we find accumulation to act as a limiting factor on the formation of lakes in response to melt, introducing new frozen material to the surface, thus cooling and storing meltwater. We find accumulation's ability to limit melt to be further enhanced by its build-up, increasing available firn air content, and thus meltwater storage capacity. Our findings are supported by comparative analysis, showing years such as 1989 to have 55% less melt, but similar lake coverage to 2020. Finally, we find that climate projections suggest future temperature increases, but steady snowfall in this region. Thus, in future there will be a greater propensity for higher lake densities on North George VI ice shelf, and associated risk of instability.

## 1 Introduction

The Antarctic Peninsula (AP) is covered by half a million square kilometers of ice and is fringed by numerous floating ice shelves (Birschandler, 2006; Davies et al., 2012). Between 1950-2004, the AP warmed by ~2.5°C, making it the most rapidly warming region in the Southern Hemisphere (Turner et al., 2005). This period coincided with extensive ice shelf thinning (Adusumili et al., 2017; Shepherd et al., 2010; Scambos et al., 2004) and collapse (Banwell et al., 2013; Rebesco et al., 2014; Scambos et al., 2000). More recently, there has been a cooling trend (-0.47°C decade[-1] between 1998-2014), demonstrating two distinct climatic epochs over the past 40 years (Turner et al., 2014; Yao et al., 2016). Pre-1998 warming has been attributed in-part to the Southern Annular Mode (SAM), a cyclic climate driver in the Southern Hemisphere estimated to have been responsible for ~30% of the change in annual near-surface air temperatures (SATs) on the AP (Marshall & Bracegirdle, 2014). SATs have, in the past, been considered an important control upon ice shelf stability, potentially causing



'significant progressive retreat or total loss' (Morris & Vaughan, 2003). Furthermore, thanks to the consideration of SAM we look at warm, dry föhn winds, a major component of the metric's impacts on the AP, causing localised melting that can cause
instability in vulnerable ice shelves (Laffin et al., 2019).

A major mechanism of ice loss on the AP is ice shelf collapse. Since 2000, both the Larsen B (LBIS) and Wilkins (WIS) ice shelves have experienced collapse events (Braun & Humbert, 2009). Ice shelves exert a buttressing effect on their upstream glaciers, limiting flow and discharge of ice into the ocean (Dupont & Alley, 2005; Glasser et al., 2011). Since 2002, reduced buttressing from ice shelf collapse has led to the loss of ~9 Gt a$^{-1}$ of grounded ice, representing one third of the total
ice loss from the AP, and its associated sea level contribution since then (Berthier, Scambos & Shuman, 2012). Supraglacial lakes (SGLs) are considered a potential precursor for ice shelf collapse, largely because widespread lake drainage was observed prior to the break-up of LBIS (Banwell et al., 2013; Glasser & Scambos, 2008; Scambos et al., 2009). Ice shelf instability is thought to be induced by hydrofracture, whereby increased hydrostatic pressure from surface meltwater propagates fractures in the ice shelf (Robel & Banwell, 2019). Repeated loading and unloading of the ice shelf from lake filling and drainage
(Banwell et al., 2019) may also induce fractures around lake basins. Ultimately, a network of intersecting fractures may result in ice shelf disintegration (Banwell & MacAyeal, 2015).

George VI ice shelf (GVIIS) is the largest ice shelf on the Western AP, occupying an area of ~24,000 km$^2$ between the mainland ice sheet and Alexander Island (Figure 1). GVIIS buttresses the flow of grounded ice from substantial sectors of Palmer Land and Alexander Island, and thus its collapse is projected to contribute ~8 mm to sea level rise post-failure
(Schannwell et al., 2018); the same as the contribution from the entire Greenland ice sheet between 1992 and 2012 (Shepherd et al., 2012). Being constrained by two landmasses however, the susceptibility of GVIIS to collapse is uncertain, given that its compressive flow regime may potentially increase structural resilience (Hambrey et al., 2015; Holt et al., 2013, Lai et al, 2020). As such, the northern GVIIS is likely much more robust to factors of ice shelf collapse, thanks to its compressive flow regime, and therefore could withstand greater forcing from SGLs versus neighbouring shelves such as Larsen B and the Wilkins ice
shelf.

The austral summer of 2020 saw extreme melting across this region of Antarctica, including the hottest day ever recorded on the continent (Robinson et al., 2020). This melting led to extensive SGLs on north GVIIS which, at their peak, covered a similar percentage of the ice shelf surface to that observed on LBIS prior to collapse (Banwell et al., 2021). Although melting was at a record high and lakes were widespread, it remains unclear whether lake coverage in 2020 was unprecedented,
and consequently whether this represented a notable increase in GVIIS's vulnerability. Here, we analyze satellite imagery acquired between 1973-2020 to assess the long-term prevalence of lakes on the northern GVIIS and provide context for the 2020 observations, considering the similarities between the high lake coverage in 2020, and of previous years with similar extensive lake formation such as 1989. In combination with Regional Climate Model (RCM) simulations, we assess the climatic drivers of lake formation and explore associated implications for future lake coverage in this region.



## 2 Data and Methods

### 2.1 Satellite Data

We used all available (< 40% cloud coverage) Landsat-1-8 (LS1-8) and Sentinel-2 (S2) satellite imagery acquired between 1973-2020 to produce 72 composite image snapshots of lake coverage on the northern GVIIS during summer months (December to March) (Table S1). This record comprises sporadic temporal coverage between 1973 and 1999 (January 1989-1991), and monthly summertime coverage thereafter (December through March 2000-2020). Where multiple images were acquired in a single calendar month, the image with least cloud cover over the northern GVIIS was used. Where multiple low-cloud images existed, we used the image acquired closest to the middle of the month. These images were merged into a mosaic to create a cloud free composite image for each monthly snapshot for the whole study region. With this composite method, only 1989 had notable, but thin, cloud cover, which was remedied with manual editing. We focused on the northern sector of north GVIIS, due to its high concentration of lakes. To map SGLs, we used a Green-Near InfraRed (NIR) NDWI (Normalized Difference Water Index) method (Equation 1) to identify water pixels within each satellite image. Many other methodologies of NDWI have been used, including multiple-algorithm approaches such as within Moussavi et al., (2020), however due to the major manual post processing component of this work, more sophisticated methods were not deemed necessary. Of the most used SGL identification algorithms (Box & Ski, 2007; Williamson et al., 2017) the Green and NIR method was found to yield the highest accuracy in our study, with the fewest false positives and negatives in testing (Text S2.1, Table S3).

$$NDWI = \frac{(Green-NIR)}{(Green+NIR)} \qquad \textbf{(1)}$$

Where Green and NIR refer to the reflectance of bands 3 and 8 from S2, bands 3 and 5 from LS8, bands 2 and 4 from LS-4-7 and bands 4 and 6 from LS-1. We applied a threshold of 0.28 to the NDWI image from LS1 and a threshold of 0.3 for all other Landsat and Sentinel variants to yield a binary lake/not-lake raster. A different threshold was used for LS1 because bands 4 and 6 in LS1 do not directly correspond to bands measured by the other Landsat instruments. Each binary image was converted to polygons, which were dissolved to produce monthly maps of SGL distribution. All polygons under 1800 m$^2$ (equivalent to two LS8 pixels) were removed (Stokes et al., 2019) and lake count, area and coverage of the northern GVIIS were calculated. To ensure a robust dataset, major manual post-processing was performed, with any misclassified clouds, bedrock, crevasses and calving fronts removed after comparison with true-color images. LS7 required further manual quality control due to the LS7 scanline corrector failure (Ihlen & Zanter, 2019). Data retention after to the scanline corrector failure is given by LS7 documentation as 78% (Ihlen & Zanter, 2019), leading to a loss of 22% of data per image. Official values of data loss were verified by comparing a series of images from January 2009, 2007 and 2005 versus imagery from contemporary satellites in 2020 and 2018. A composite image was made for each period and the area not covered by LS7 imagery was quantified, producing an average retention of 77.7%, within margin of error of the 78% value given in Ihlen & Zanter (2019). Manual quality control for LS7 data involved calculating quantity of data lost due to scanlines (22%) and reducing all non-LS7 values





by this amount for all data to be comparable (Text S2.2, Table S4). In total the data acquired between 2000-2010 had 9 missing months (most typically at the end of the melt season) in the record, and the pre-2000 LS1-5 record produced viable imagery

in only 1973 and 1989-1991. Translucent cloud cover was present on GVI in LS4-5 images acquired in 1989-1991, which made automatic lake detection using NDWI difficult. Therefore, lakes beneath clouds in these images were delineated manually. Finally, LS1 imagery in 1973 was georeferenced manually to correct an apparent geolocation error of ~15 km, before lakes in this image were delineated.

## 2.2 Climate and Firn Data

To assess the role of climatic forcing on SGL evolution, we analyzed monthly climatic data including the SAM index (Marshall, 2003), 2-meter hourly SATs (averaged into monthly means) from the Fossil Bluff Automated Weather Station (AWS) (UK Polar Data Centre, 2020); and ice surface temperature and monthly melt (meters of water equivalent, m.w.e.) derived from the Modèle Atmosphérique Régional (MAR) (Agosta et al., 2019, Kittel et al., 2021). MAR, which was forced at its boundaries by ERA-Interim reanalysis (Dee et al., 2011), was selected over other available RCM products as it offered

the closest agreement to field-based temperature measurements recorded by the AWS (Text S5) and a resolution of 7.5 km. Uncertainty in the MAR model was considered in its selection, taking into account points from Agosta et al., (2019) and Mottram et al., (2020). However, as our primary analysis focuses on lake coverage, and comparison to RCM for possible correlations, we limit our consideration of these uncertainties to prior literature. We also modelled the evolution of north GVIIS's firn layer using the $Ar_{MAP}$ parameterization in the Community Firn Model (CFM; Stevens et al., 2020; Verjans et al.,

2020), to derive monthly estimates of ice lens depth, refreezing, runoff and firn air content (FAC) in the upper 10 m of the snowpack, with FAC representing the pore space available for meltwater storage. We use $Ar_{MAP}$ due to its relevance to the study region, as it is a reparameterization of the firn densification model developed by Arthern et al. (2010) which uses a Bayesian calibration framework on a dataset of firn cores from both Antarctica and Greenland, as detailed in Verjans et al., (2020). The CFM was forced with daily data from MAR between 1979-2020, specifically surface temperature, surface melting

and precipitation for all grid cells within our study area. To evaluate the future evolution of the climate of north GVIIS, simulations were performed with MAR forced by three general circulation models (GCMs) - CESM2, CNRM-CM6 and ACCESS1.3 to explore projected changes in air temperature and snowfall, up to 2100 using their respective high-emission scenario (ie., RCP8.5 for ACCESS1.3 and SSP585 for CESM2 and CNRM-CM6).

## 2.3 Statistical Analysis

To analyze statistical correlations between climatic processes and lake presence, a multivariate regression analysis was carried out using climatic and CFM variables as predictors for lake coverage. We used the backwards selection method, where all potentially relevant variables from the CFM, MAR, and the SATs recorded at the Fossil Bluff AWS were included in a multivariate regression model (Text S6). These included air and ice surface (skin) temperatures, snowfall, FAC, runoff, melt, ice lens depth and refreezing. Variables were then removed procedurally, with the most non-significant variable being





removed with each iteration, until a statistically significant fit to the data was found, and the adjusted $R^2$ decreased with the removal of further variables. The statistic used for comparing outputs was the adjusted $R^2$ value rather than $R^2$ as adjusted $R^2$ penalizes regressions with large numbers of variables, thus minimizing noise fitting. Due to many of the predictor variables displaying multicollinearity (e.g. SAT is a direct influence on melting), we were unable to draw meaningful conclusions connecting lake coverage to climate variables from this analysis and hence used the regression to identify variables for analysis

via univariate correlation.. Further study was carried out in the form of univariate regression analyses, using the multivariate analysis as a basis. Each variable was regressed versus lake coverage, with the most significant being regressed versus melt, and as ratios with melt versus lake coverage. This was done to ensure each result's contribution to lake coverage was significant, due to the lack of effectiveness of the multivariate method. Standard deviations were also calculated in order to define exceptional variance from the mean and thus to identify exceptional years of lake coverage versus controlling variables.

**3 Results & Discussion**

**3.1 Lake Coverage 1973-2020**

We produced monthly assessments of SGLs on the northern GVIIS for each melt season (December-March) after 2000, and annual (mid-January) estimates for 1973, 1989, 1990 and 1991 (Figure 1). The seasonal maximum in lake coverage is consistently observed in January or February, coinciding with the peak of the AP melt season (Smith et al., 2007). We find

that in the record melt year of 2020 (Banwell et al., 2021), 11.8% of the study region was covered by lakes, 1.93 standard deviations (SD) higher than the long-term mean of 4.1%, and greater than the ~10% coverage of lakes on LBIS prior to its collapse (Banwell et al., 2013). Of course, this does not necessarily mean north GVIIS is surpassing its limit of vulnerability, due to previously mentioned structural factors in the ice shelf's compressive flow regime. However, although lake coverage in 2020 was well above average, our new long-term record shows that it was not unprecedented, and indeed that there was

similar coverage (12.1%) of lakes 31 years prior, in the summer of 1989 (Figure 1). However, due to inherent uncertainty in lake delineation, the difference between coverage values in 1989 and 2020 is not statistically significant, and thus both years show very similar coverage totals.



**Figure 1.** Supraglacial lakes on north GVIIS in years where there is extensive lake coverage. (a) Sentinel-2 image acquired on 19 January 2020. Inset shows the location of the study area on the AP. Landsat-4-5 images acquired (b) 4 February 1991, (c) 15 January 1990, (d) 28 January 1989, and (e) Landsat-1 image acquired on 9 January 1973. All images are RGB true color composite, except (e). The black outline identifies the study area extent. Red, yellow and blue bars in (b)-(d) are satellite band edges.

Between 2000-2020, interannual variability in lake coverage is high, with no significant trend (p = 0.61; t-stat = 0.52). However, within this period we see two distinct epochs. Between 2000-2010 we see a minor increasing trend in lake area (0.7% increase per year), whereas in the subsequent years there is no significant trend. From 2010-2011 lake coverage decreased by 5.48%, and remained low until 2020. Throughout the 24 years for which we have available imagery, we find 6 high-coverage years, quantified as when lake coverage is >1 SD (4%) above the mean; namely 1989 (12.1%), 2020 (11.8%), 1990 (11.5%), 2010 (9.2%), 1973 (8.9%) and 1991 (8.5%). Whilst the earliest date for which we can quantify lake coverage is 1973, it is important to note that lakes have been observed on GVIIS by field studies as early as 1936 (Stephenson & Fleming, 1940; Reynolds, 1981). Thus, GVIIS has hosted SGLs for close to a century at least.

We find spatial variability in lake extent and morphology throughout the study period, on both regional and local scales. Lakes typically form around the grounding lines in most years, and cover the entire width of north GVIIS in years





where lake coverage is high. The Western grounding line in the proximity of Ablation Point (Figure 1b) is the only region where lakes are present in every year of our record. In general, lakes towards the south of the study area are larger and

interconnected, whereas lakes north of ~71°S are discrete (Figure 2). This pattern is most apparent in years where lakes are most extensive, namely in 2020 and 1989. Furthermore, we observe that the spatial distribution of lakes has evolved through time, with lakes being more concentrated to the south in 1989 (Figure 2d), whereas in 2020 lakes are more uniformly distributed (Figure 2b). In 1989, for example, lakes covered 192.7 km$^2$ (9.1%) of the area south of 71.8°S, whereas in 2020, lakes covered less than half of this area (74.2 km$^2$, or 3.7%).




**Figure 2.** Supraglacial lake extent in the high lake years of (a) 2020 (pink) and (c) 1989 (green). The inset histograms (b) and (d) show lake coverage per 0.05 degrees latitude from North to South. The red outline denotes the study area. The white background shows grounded ice, while pale blue indicates the floating ice shelf.

### 3.2 Climatic controls on lake coverage

We used our record to explore the relationship between climate forcing, firn air content and lake formation, by identifying those variables which had the highest predictive power with respect to lake coverage (defined as $|r| => 0.45$). The variables that met this requirement were melt, SAT, SAM magnitude and summer accumulation (Figure 3).

Climatically, we find 2020 to be an unprecedented year in terms of SATs and melt with mean values reaching -1.2°C and 10.9 m.w.e respectively (respectively 3SD and 3.5SD from mean) while accumulation values are within expected variance

at 0.86 SD below mean. In-situ records from the nearby AWS show 2020 to have the highest recorded mean summer SAT since records began (Figure 3b), deviating by 2.9 SD from the mean. This finding links closely to other studies which found 2019-2020 to have longer continuous periods of above 0°C SATs than any other previous melt season (Banwell et al., 2020). For comparison, simulated total summer melt for the same year from MAR is 3.4 SD above the mean. Hence, it is interesting to observe that both 1990 and 1989 exhibited similar lake coverage to 2020, despite 1989-1990 experiencing cooler monthly

mean temperatures at the January peak of the melt season than 2020 (-1.4°C and -1.6°C cooler, respectively). The same is true for melt in these years, with 1990 experiencing 44.6% less melt, and 1989 experiencing 55.3% less melt than in 2020, and yet only showing 0.3% more lakes. Hence, there must be factors limiting the response of lake coverage to melt in 2020, or factors enhancing the lake coverage of 1989.







**Figure 3.** Timeseries of climatic and firn variables versus lake coverage (a-e). Reported lake coverage values are the maximum for each summer melt season and shown as dashed black lines. Winter accumulation values correspond to those recorded in the preceding winter. Climatic variables (panels a and b) are drawn as dotted lines in periods where climate data are available but no lake data were acquired. Correlation values and scatterplots available in supplementary figure S7.

Of all the predictor variables considered within our regression analysis, melt (figure 3a) exhibits the strongest correlation with lake coverage (r = 0.63). This is unsurprising, given that increased melt leads to greater water availability for lake formation within the surface hydrological system. However, melt does not explain all of the variance seen within our lake coverage dataset, with an $R^2$ value of 0.39, i.e. melt only explains just over a third of the variance in lake coverage. This indicates that there is not a simple linear relationship between melt intensity and lake formation. Furthermore, while 1989 shows very similar lake coverage, melt experienced in this year was 55% less than in 2020. Due to only a portion of variance being explained by melt, there may be additional factors which explain the variance, as despite large variance in melt, we see little variance in coverage between the two highest peaks (55% difference in melt 1989-2020). Hence, below we consider other climatic factors that may be related.

It is also unsurprising that summer SAT (figure 3b) acts as a control on SGL coverage (r = 0.52). Summer SAT having the correlates strongly is important since SAT directly influences both melting and the evolution of the firn pack (Dell et al., 2020) (Figure 3b). This likely explains why several of the years prior to 2000 exhibit high lake coverage, as the decadal mean summer SAT was 0.7°C higher than it was post-2000. During 1989-1990, when lake densities were similar to those observed in 2020, seasonally-averaged summer SAT's were 1-1.2°C higher than the pre-2000 mean at -2.6°C.

We also find that SAM magnitude (figure 3c) anticorrelates moderately with lake coverage (r = -0.47), with high lake coverage typically occurring in years where the annual mean SAM is close to zero (Figure 3c). Whilst SAM is a broad-scale control on SAT in the southern hemisphere, SAM also controls wider climatic variables including oceanic currents and winds (Marshall & Bracegirdle, 2015). Within this context, the latter is important because the presence of föhn winds on the AP are known to enhance surface melting (Cape et al., 2015; Laffin et al., 2019). Years of extreme SAM magnitude (>1) typically show low SGL coverage. This may be because in years with high positive SAM, westerly winds are stronger and can carry weather systems over the mountains to the eastern AP, decreasing temperatures and humidity due to build-up of sea ice against the west AP, while high negative SAM tends to be associated with dominant high pressure systems and weaker winds, thus decreasing the effect of föhn winds (Marshall et al., 2006). With low intensity SAM magnitude (<1), warmer and wetter conditions prevail, leading to increased melt and precipitation on the western AP (Marshall, 2003; Marshall et al., 2006). Overall, the sensitivity of SGLs to SAM is likely to be a result of a complex combination of factors, including snowfall and melt from the SAM metric.

Summer accumulation (figure 3e) is strongly anticorrelated with melt (r = -0.61), and to a lesser degree with lake coverage (r = -0.35). The former relationship is likely to hold because high summer accumulation will typically occur in years with more cloud and vice-versa; insolation will therefore be higher in years with less cloud, increasing energy available for melt. The latter is either due to increased accumulation providing more storage space for meltwater within the firn layer, preventing it from ponding in lakes, or because increased accumulation can lead to the burial of supraglacial lakes (Lenaerts





et al., 2017). Winter accumulation and increased November FAC (figure 3d, e) also contribute towards a dampening effect, limiting the response of lake coverage to lake enhancing variables such as melt and SATs (Alley et al., 2018; Kuipers Munneke, 2014) and indeed we find that November FAC is anticorrelated with lake coverage (r = -0.2; p = <0.01). If 2020 is excluded due to its high leverage as a result of its exceptional coverage versus climate forcing, the strength of this anticorrelation becomes greater (r = -0.25). Winter accumulation correlates strongly with November FAC (r = 0.57). This suggests that while

November FAC is strongly dependent on winter accumulation, the relationship is non-linear, and indeed it is likely that cumulative buildup of FAC from consecutive low melt/ high snow years is also important. We also find a relationship between winter accumulation and melt (r = 0.32). These relationships suggest that FAC dampening on lake coverage response may be underestimated using just the correlation between winter accumulation and lake coverage. We therefore suggest that the buildup of FAC in the 2010s from low melt and high accumulation (figure 3a, e), dampened the lake coverage response to the

exceptionally high melt which occurred in 2020; where we see exceptionally high melt. This is supported by the observed differences in morphology of lakes in 2020 (where lakes appeared more discrete), and 1989 (where lakes appear to be more connected) suggesting that the water table within the firn was higher in 1989. The buildup of FAC pre-2020 is suggested by the lack of lake presence in the preceding years; albeit a lack of available imagery prevents us from making a similar observation in the years preceding 1989, and between 1991-2000.

**3.3 Future Climate Simulation**

To assess the potential impact of future changes in climate on north GVIIS, we examine projections of snowfall, rainfall and SAT simulated by MAR (Kittel et al., 2021) across the AP (Figure 4). These projections suggest that air temperature will increase on north GVIIS at a rate of ~0.5°C decade$^{-1}$ between 2020 and 2100 (Figure 4b). In contrast, snowfall is projected to remain constant throughout this period (Figure 4a). Although precipitation typically increases in warmer

climates, MAR suggests that over north GVIIS the increase will favor rain over snow, with total annual rainfall on north GVIIS predicted to increase by 1.81 mm w.e. a$^{-1}$ annually from 1980-2100. Thus, it is likely that the dampening effect of snowfall-driven increases in FAC that we propose moderated 2020's lake extent will diminish in future, as the precipitation phase shifts from snow, which increases FAC, to rain, which does not. This pattern is also apparent on other AP ice shelves (Figure 4), suggesting that lakes are likely to become more widespread across AP ice shelves in the future, as well as on north GVIIS in

particular.





**Figure 4.** Yearly timeseries at a representative MAR grid cell on the northern GVIIS using MAR forced by three GCMs (1980-2100) and compared to MAR forced by ERA-Interim (1980-2020) for (a) projected total annual snowfall and (b) projected mean SAT. The black line in (a) and (b) is the mean of the GCM forced model iterations. Map of the rate of change ($\delta$) in (c) SAT, (d) total annual rainfall, and (e) total annual snowfall across the AP (1980-2100). Rainfall and snowfall are given as millimeters water equivalent (mm w.e.). The red points in panels c-e show the location of the MAR grid cell from which the timeseries in panels a and b were computed. Anomalous band in top right of (d) and (e) is due to boundary conditions of MAR being applied over a buffer zone close to the RCM edge.



## 4 Conclusions

We use optical satellite imagery to study SGL coverage on north GVIIS between 1973-2020. We find no significant trend in lake coverage over that time, although high lake densities were more common in the 1989-1991 period than since 2000. The highest lake coverage is found in 1989, 1990 and 2020. In all three years, lake coverage was greater than 10%, similar to that observed on the LBIS immediately prior to its collapse in 2002 (Banwell et al., 2013; Leeson et al., 2020). Despite this, these years show differing climatic conditions, with 2020 exhibiting the highest melt on record, while 1989-1990

remained closer to the climatic mean. This suggests that high melting alone is not the only factor influencing lake coverage on the northern GVIIS.

     Based upon statistical analysis, we find that the temporal variance in lake coverage on the northern GVIIS is best explained using summer melt, SAT, annual SAM magnitude and a combination of accumulation and November FAC. Generally lower temperatures and melting between 2000-2019 facilitated an increase in FAC and correspondingly lower lake

densities were observed than in the preceding period 1989-1990. 2020 was a unique case in our record because it experienced melt, SAT and SAM values favorable to unprecedented lake formation, yet experienced similar lake coverage to 1989. We find that the build-up of firn from successive low lake seasons since 2013 contributed to the dampened response to climate forcing we see in 2020. Indeed, it is likely that had FAC not built up since the early 2010s, 2020 may have exhibited both the highest melt and lake coverage on record; albeit without equivalent satellite imagery from the years preceding 1989, this cannot

be determined unequivocally.

     Projections made using MAR for the Antarctic Peninsula suggest that air temperature and rainfall will increase between 2020-2100, but snowfall will remain at roughly present-day levels. Thus, we can expect the availability of surface water to increase, causing a decrease in FAC. It is reasonable therefore to assume that lakes will become more abundant on AP ice shelves in coming years. This, together with the fact that mean SAT on the northern GVIIS is expected to consistently

exceed a proposed limit on ice shelf viability after 2040, means that understanding the structural resilience of this ice shelf to sustained hydrological forcing should now be a priority for future study.






**Acknowledgements**

We would like to thank Diarmuid Corr for assistance in manual delineation method testing used to check the viability of the Green-NIR NDWI method. We are grateful to ESA's 4D Antarctica and Digital Twin Earth (Antarctica) projects for providing financial support for this work. Mal McMillan acknowledges the support of the UK NERC Centre for Polar Observation and Modelling (grant no. cpom300001), and the Lancaster University–UKCEH Centre of Excellence in Environmental Data Science.

**Data Availability Statement**

Sentinel 2 data are available online (https://scihub.copernicus.eu/). Landsat data are available online (https://earthexplorer.usgs.gov/). Fossil Bluff automated weather station data are available online (https://data.bas.ac.uk/metadata.php?id=GB/NERC/BAS/PDC/00794).

SAM data are available online: (https://legacy.bas.ac.uk/met/gjma/sam.html).

MAR    outputs    are    available    on    Zenodo:    https://zenodo.org/record/4529004#.YLToiHUzaV4, https://zenodo.org/record/4529002#.YLToiXUzaV4, https://zenodo.org/record/4529002#.YLToHXUzaV4

**Author Contribution**

The authors are ordered by their relative contributions. TJB initiated the study, carried out lake delineation and comparison to climatic datasets and produced the manuscript draft. AAL and MM both provided major support in iterating pre-submission 310    reviews, and support throughout the project in method determination and establishment of method and discussing results. VV provided firn model data and helped interpret results. JC produced figure 4 and aided in discussion of future climate. CK provided MAR data and information relating to its interpretation. All authors were involved in discussing results and editing of the manuscript.

**Competing Interests**

The authors declare there are no conflicts of interest in this work.

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
