# Peer review of "Changes in Supraglacial Lakes on George VI Ice Shelf, Antarctic Peninsula: 1973-2020"

_The Cryosphere, 2021_

## Referee Comment (RC1)

REVIEW OF "Changes in Supraglacial Lakes on George VI Ice Shelf, Antarctic Peninsula: 1973-2020"

T. J. Barnes, A. A. Leeson, M. McMillan, V. Verjans,  J. Carter, C. Kittel

General assessment:

The authors of this paper attempt to synthesize the satellite imagery record, local weather station data, firn model output, climate reanalysis model output, and CMIP future climate model output in order to better understand the future expansion of the George VI's surface meltwater drainage system. This is motivated by the recent findings of unprecedented melt in the 2019-2020 melt season. The motivations behind this study,  to use the available data to give an assessment of present-day conditions of the surface drainage system to better predict its future evolution, is well-thought out, valuable, and promising for future study. Other studies have done similar multi-year assessments of Antarctic ice shelf surface hydrology systems (Langley et al., 2016, Stokes et al., 2020, Spergel et al., 2021). However, the authors present many different means of showing a disconnect between climate and surface conditions with lake coverage, but do not discuss that lake coverage, i.e where melt is observed ponding in satellite imagery, is mainly controlled by surface topography. The pre-existing surface depressions (discussed in Reynolds, 1981) must be filled to overspilling before water can drain over the surface into new areas/depressions.  I am not familiar with the surface topography of GVIIS, but other ice shelves have more-or-less U-shaped depression cross-sections, so the addition of more meltwater does not change the surface area of the water body. It is only when all available space is filled with water that the surface area of water coverage expands via over-surface drainage. What the authors seem to describe with their analysis of similar meltwater lake coverage between 1989 and 2020 is not a dampened response to climatic forcing, but the fact that meltwater pond coverage increases as water flows and partially fills depressions nearby to meltwater production, but meltwater coverage plateaus as the partially-filled depressions fill, and only once overspilling occurs does lake coverage increase again significantly.

I would recommend refocusing this paper on the changes observed in melt pond distribution on GVIIS between the 1980s, as described in Reynolds, 1981, and where melt is observed today. There is a lot of value to giving a base-line and a detailed description of the inter-annual variability in the ice shelf's hydrology. I would recommend a thorough search of the literature to give a broader context to the authors' findings on GVIIS.

Grades are from 1 (excellent) to 4 (poor). Your MS is graded as follows::

Originality: 2

The purpose of the paper, to assess the decadal trends in a persistent surface drainage network has a lot of merit.

Scientific Quality: 4

The oversight of topography controlling where melt forms, and what that means for measuring meltwater lake coverage with satellite imagery, makes a lot of the analysis done in the paper unsuccessful in proving any climate-surface hydrology mechanisms.

I also have many questions about methods that are not addressed in the paper. The results are reliant on the threshold of NDWI, the comparison of imagery coverage across the years, and uncertainty in manual mapping. These three issues and the uncertainty they contribute should be discussed.

Significance: 4
As the paper is now, the points that are presented successfully (persistent, widespread melt on GVIIS; inter-annual variability in melt production leads to variability in meltwater lake coverage; meltwater being divided between ponds and firn pore spaces) are not novel enough to be significant. In its current form, the paper is unsuccessful in supporting a new mechanism for climate-surface hydrology interaction, the "dampening effect" of increased firn air content on lake coverage.

Presentation Quality: 3
Much of the paper is well written, but there are a few issues in the paper's presentation: 1) there remains a number of passages that could use scientific, quantitative terminology instead of conversational. 2) quantities such as averages, sums, etc, should be precise in what they are describing to avoid ambiguity. 3) The figures should be revised to be more readable, especially the time-series plots. 4) Much of the material presented in the supplementary materials is critical to assessing the paper, and should be brought into the main text.

**Main Comments:**

1. The paper needs to be reassessed after considering how surface topography affects where water pools. Several of the proposed mechanisms and causal relationships between climate, firn, and meltwater lake coverage need to be reconsidered, and revised if still true or removed if no longer true.
2. The methods by which lake pixels are selected need to be further explained. Moussavi et al. (2020) would be a good reference if the new method is to be kept, but I would recommend using Moussavi et al.'s available code for Landsat 8 imagery and discuss the process used to select the NDWI thresholds for Landsat 1-7. I also don't understand what the scaling of lake pixels derived from non-Landsat 7 imagery includes, but the uncertainty introduced by this needs to be discussed.
3. Many of the assertions about climate effects on meltwater lake coverage presented in the discussion/conclusion need to be supported by data or citation of the literature. The choice of MAR is discussed in the supplementary materials, but the authors also seem to use MAR output as a single point rather than a spatially-varying raster dataset.

4. Some sections need to be rewritten to clear up ambiguity in what was done, what is being extrapolated, etc.

Line by Line Edits:

L15. "build up strong" > "build up. Strong..."
L15. I think "precede" or "yield" is meant here instead of "entail."

L26. Adusumilli not Adusumili

L32. Consider "In considering SAM, we also analyze..."  Instead of "Furthermore, thanks to..."

L33. What metric is this being referring to?

L38. "reduced buttressing due to ice shelf collapse is estimated to have led to.."

L40. "a theory supported by widespread lake drainage observed prior..."

L46. In George VI Ice Shelf, "Ice Shelf" should be capitalized

L47. The area number needs to be cited.

L55. Hottest day means highest daily average temperature recorded?

L57. What are the values of the two percentages being compared?

L59. Vulnerability - "vulnerability to meltwater-driven hydrofracture."

L72. How were the individual satellite images merged? In what software?
From Table S1, it looks like there are several different Landsat rows used: did the authors have a change in coverage of the area of interest depending on which row was used?

L75. In what software was this lake pixel identification conducted?

L77. I suppose I don't understand why Moussavi et al.'s code was not used, because it has published results and an extensive workflow to remove rocks, clouds, shadows, and other false positives.

L84. Where did the thresholds the study is using come from?

L89. Are shadows also removed during this pre-processing step?

L90. Is the removal all manual or automated?

L96. See Main Comment 2

L99. GVIIS

L110. What does it mean that the authors took into account points from Agosta et al. 2019?

L112. I think the authors mean that the limitations of MAR are acknowledged, but an analysis of these limitations is outside of the scope of this paper. This section should be moved to the discussion. It is not appropriate to include it in the data and methods section.

L113. What is the resolution, temporal and spatial of ArMAP? Have others tested it compared to ground truthed data nearby?

L115. I don't understand this sentence. What about ArMAP is connected to the study region? That it was created from Antarctic firn cores?

L122. Citation required for CMIP models.

L127. I think the authors should include any pertinent information from S6 here in the main text.

L132. 1. This should be in the discussion, not methods. 2. If this multi-variate method did not yield useful results, why is it included in the paper? Can the authors not just present the univariate correlations?

L136. "ratios with melt versus lake coverage." - What are the ratios?

L137. This is not a necessary statement: "Standard deviations were also calculated in order to define exceptional variance from the mean and thus to identify exceptional years of lake coverage versus controlling variables."

L143. "the previously reported peak of AP melt season"

L146. This does not belong in the initial introduction of the results. Also, try to adopt a more objective tone: "Of course" comes off as too conversational.

L150. It would be better to present the actual uncertainty in areal lake coverage in 2020 and 1989.

L151. lake coverage. There's not a total, unless I am missing a summation?

L156. Is there a figure associated with this results section? If not, one should be added.

L156. There is no significant linear trend, right?

L161. I believe "defined" is meant, not "quantified"

L161. "mean percent coverage"

L161. I would split this into a separate sentence. "We find the following years to have high lake coverage: ..."

L162. This statement should be in the introduction
L164. Reynolds, 1981 is missing from the bibliography.

L165. I don't understand what morphology means in this context. Shape of lake margins? Location?

L166. I don't know what this means: could the authors say from eastern to western grounding lines or from the northern to southern regions of our region of interest?

L167. western should not be capitalized

L167. "near" instead of "in the proximity of"

L168. Considering there was a decently in-depth study of why George 6's lakes form in particular areas (Reynolds, 1981), there should be more spatial information available to qualify and give more information to this finding.

L169. "form interconnected networks across surface depressions"

L169. Instead of "discrete" - "do not show the same connectivity between surface depressions".

L170. "such as" not "namely"

L170. Two points do not show evolution. One can describe that two different spatial distributions were observed in these two composite maps, but unless the authors see a graduated shift between these two different distributions, the authors can't extrapolate a time-varying process.

L171. reverse the order that (d) and (b) are presented.

L172. I don't know how the 9.1% was calculated.

L174. I would like to also see a description of lake area histograms during each period.

L179. There should be a discussion of spatial variability in temperature considering the temperature data is a raster not a point.

L181. "Melt" is MAR modelled melt production?

L181. surface temperatures as reported at the AWS or those from MAR?

L181. Accumulation always refers to snow accumulation?

L182. Is this the summer (DJF) of 2019-2020?

L182. This is the annual mean temperature?  I find that number very difficult to believe.

L183. Again, is this daily mean snow accumulation or annual snow accumulation?

L183. total accumulation over 2019-2020? daily output?

L188. Are you describing a single monthly average 2m air temperature? From the AWS? If this is a key point, it should be clear where this temperature is from.

L192. is this the number of lakes or lake coverage? In either case, this is controlled by both pre-existing surface topography and by melt input. See Main Point 1.

L207. With this and other r values in the text, a p-value is required to show that the correlation is statistically significant.

L212. I wonder if correlating the specifics of SAM to melting on GVIIS is beyond the scope of this paper.

L225. high cloud cover could also increase the trapping of long-wave heat.

L231. Why is 2020 being excluded? It's an outlier?

L236.  I'm having trouble following the logic here around the dampening mechanism. Reword possibly?

L236. There's a lot of supposition behind the dampening effect the authors are describing. I worry that this dampening effect is the result of pre-existing surface topography needing to be filled to overspilling in order to increase lake coverage. In any case, the build-up of firn air content is not described previously.

L250. I believe that firn air content did moderate over-surface drainage extension, but rain will also deepen the depressions on the ice, which will also 'dampen'/slow the expansion of water into new depressions.

L265. I don't think the authors presented lake densities in 1990 or 1991

L275. "We find that the build-up of firn from successive low lake seasons since 2013 contributed to the dampened response to climate forcing we see in 2020" - This point needs the presentation of more data previously in the paper.

L282. This presupposes that the available accommodation space in the existing depressions will be filled to over-spilling.

L283. Where is this statement on ice shelf viability from?

Figures.

Figure 1.
- labelled points and insets should be on the same panel, (a) in this case.
- My reading order of this figure is left to right, top to bottom. I would rearrange the panels to fit this reading order.
- I'd rather see one satellite image with labelled locations, and then maps of identified lakes in the remaining panels. I can't get much information from these satellite images presented like this.
- I would make the area of interest's outline thicker to differentiate it from the lat/lon gridlines
- what is e if it isn't an RGB true color composite?

Figure 2.
- What images/dates are these two maps composed of?
- I would put the legend in a box to the right of the figure rather than at the bottom of (a).

Figure 3.
- This figure is very difficult to parse. Have you considered making a scatterplot with each point labelled by year, like figure S7?
- Why is there a gap in the November FAC and accumulation time series?

Figure 4.
- I can't read any sort of trend in plot A.
- These maps should be focused on only the area being studied.
- What is $\partial T$ here? Difference in annual mean? In summer mean?

Supplemental Material:

Table S3's minimum area column is unnecessary.

S53. I don't understand, the authors decreased the total area in images with more coverage?

S100. This shouldn't be in supplemental methods descriptions, but rather in the main text

---

## Referee Comment (RC2)

**Anonymous Review of 'Changes in Supraglacial Lakes on George VI Ice Shelf, Antarctic Peninsula: 1973-2020'. Barnes et al. (2021)**

**Overview**

This manuscript presents a record of surface melt area on George VI Ice Shelf between 1973 and 2020. It identifies high melt years in 2020, 1990, and 1989. The authors map lake areas using an NDWI green and infrared method, alongside manual delineation of some scenes, and heavy manual post-processing. They then consider the patterns in surface meltwater areas in the context of climate and firn data, in addition to using CMIP to consider how GVI Ice Shelf's meltwater system may evolve into the future.

Overall, it is clear that a lot of work, comprising many data sets, has been put into this manuscript. However, the methodological steps are hard to follow, and the significance of the findings are therefore hard to evaluate with certainty. I recommend that the author's re-write the paper in many areas, and focus on presenting the methods and results with greater clarity, taking into account my suggestions below.

**Novelty**

Whilst other work (e.g. Banwell et al., 2021) has discussed melt extents on the GVIIS, this work adds some new insights through the use of climate (MAR) and FAC data. However, these additions need to be presented more clearly for the novelty to be fully conveyed.

**Scientific Quality**

The main text fails to clearly communicate the methodological steps taken, and does not account for error, particularly when delineating lake areas using both NDWI and manual methods. This needs to be improved.

**Significance**

The colloquial tone of the manuscript makes the significance of the work hard to assess, and the science is not yet fully convincing. This needs to be addressed. However, the discussions on firn air content and future climate modelling of GVIIS is very interesting, and if the paper can be re-written to a higher standard, these points may be better conveyed.

**Presentation Quality**

The presentation quality needs to be addressed in order for this paper to be published. The paper often has a very colloquial tone, which rambles, and therefore much of the text is

hard to follow. The methods need to be clearer throughout, and justification for methodological decisions should be provided. Generally, figures are ok, but please make sure figure labels etc are legible.

**Main Comments**

The abstract of the text needs to be re-written. When compared to the introduction it is of a lower standard of writing, and it doesn't convey the key findings well.

When describing George VI Shelf (Lines 47-55), the authors need to do some wider reading of literature. For example, they should use past work here to describe the different glaciological settings of the north and south GVIIS. A study area figure is also required.

It makes little sense to me to use the NDWI Green and Near Infrared method over the NDWI blue and red method, given that the majority of literature would use the latter, and this has been well justified in many previous papers. I am not convinced as to why the authors chose to use this alternative thresholding method, and the text in S2 still does little to convince me. It would be interesting to see some maps showing the differences between the two thresholding approaches.

The lakes in some imagery were manually delineated, yet there is no mention of the error that should be considered when comparing these manually delineated lakes to lakes found using the thresholding method. Overall, the authors should consider the errors associated with all methods, and reference these where appropriate.

The authors state that they use a different threshold for Landsat 1 because the bands do not correlate with the other Landsat instruments. But I question whether the Sentinel-2 bands correlate? If not, why did you not use a different threshold for that too?

Is there full ice shelf coverage for every data point investigated? If not, how much of the ice shelf is 'missing'?

The authors only show satellite imagery of GVIIS in maximum melt years, however they comment (Line 167) on the spatial organisation of surface meltwater in low melt years too. It would be useful to see some figures showing this, to allow the reader to see the changes that occur over time.

The authors suggest that they convert the areas for all data that wasn't affected by the Landsat-7 scan line failure, ultimately reducing the areas? This is a questionable decision as it broadly means the data presented is not representative of the true area of melt on GVIIS,

which is an important statistic to have. I suggest the authors present both the converted and unconverted data.

**Line by Line Comments**

Line 33-35: Sentence does not read well, colloquial tone.

Line 36: Change to Larsen B Ice Shelf (LBIS) and Wilkins Ice Shelf (WIS).

Line 37: Reference here (and elsewhere) is surely an example of one applicable piece of literature? If so use (e.g. ref) as opposed to just (ref).

Line 67: 'Landsat-1-8' Remove hyphens. Landsat 8 should not be hyphenated, and Sentinel-2 should be.

Line 73: 'Where multiple low cloud images'. How did you quantify low cloud images?

Line 73: How did you mosaic? Did you simply put any image preferably on top? Or the most recent image? Please specify.

Line 74: 'Remedied' is an odd word to use – change.

Line 74: If you focussed on the Northern sector then show the readers how you define this area in a study area map.

Line 129: change 'most non-significant' to least-significant

Line 135: Extra '.' – remove

Line 148: Ref for compressive flow regime?

Line 155: How was the study area extent decided? Manually? Specify in the methods.

Line 166: In this context, what do you mean by regional and local?

Line 184: Double use of 'respectively'.

Lines 203-207: This section doesn't read very well and feels repetitive. Try and condense

Line 208: 'It is also unsurprising' – too colloquial

Line 208-209: 'having the correlates' – I assume this is a typo?

Line 274: Insert comma after 'Generally'

**Figures**

**Figure 1:** If e isn't a true color composite, then what is it?

**Figure 3:** Put units in brackets

**Figure 4**: Put units in brackets.

Move X axis label 'Year' below the years.

For plots c, d, e make colour bar label font size larger.

**Supplements**

**Table S1:**
Table heading required – despite it being in the other supplementary document

Why are two Landsat 5 images highlighted in red?

If the authors have the AOI coverage for each image, they could add this information

**Text S2:** Much of this should be explained in the main text.

**Text S3:** Again, much of this should be explained in the main text.

**Figure S7:** Put units in brackets.

**References:**

Banwell, A.F., Datta, R.T., Dell, R.L., Moussavi, M., Brucker, L., Picard, G., Shuman, C.A. and Stevens, L.A., 2021. The 32-year record-high surface melt in 2019/2020 on the northern George VI Ice Shelf, Antarctic Peninsula. *The Cryosphere*, *15*(2), pp.909-925.

---

## Author Comment (AC1)

**Changes in Supraglacial lakes on George VI Ice Shelf, Antarctic Peninsula: 1973-2020, Thomas Barnes et al. Response to reviewers.**

Thanks to the reviewers for offering their time to review our manuscript, we appreciate the posted comments, as they will aid in improving our manuscript. Each of the major comments will be addressed in turn, with responses given beneath.

**Review 1**

1. The paper needs to be reassessed after considering how surface topography affects where water pools. Several of the proposed mechanisms and causal relationships between climate, firn, and meltwater lake coverage need to be reconsidered, and revised if still true or removed if no longer true.

To approach this point we intend to look at a high resolution DEM to find out whether surface topography and water pools show a connection. Based on the results of this investigation, we will identify whether or not they support the above hypothesis.

2. The methods by which lake pixels are selected need to be further explained. Moussavi et al. (2020) would be a good reference if the new method is to be kept, but I would recommend using Moussavi et al.'s available code for Landsat 8 imagery and discuss the process used to select the NDWI thresholds for Landsat 1-7. I also don't understand what the scaling of lake pixels derived from non-Landsat 7 imagery includes, but the uncertainty introduced by this needs to be discussed.

This point will be approached by comparison of the results of Banwell et al., (2021) where Moussavi et al (2020)'s methods were used to compare an alternative method to the results of this study. This was done previously through correspondence with A. Banwell, but not included in the manuscript. Additionally, reasoning behind thresholding will be discussed. The final part of this comment is addressed in the response to point (8) from Reviewer 2.

3. Many of the assertions about climate effects on meltwater lake coverage presented in the discussion/conclusion need to be supported by data or citation of the literature. The choice of MAR is discussed in the supplementary materials, but the authors also seem to use MAR output as a single point rather than a spatially-varying raster dataset.

Effort will be made to improve referencing in relation to climatic effects on lake coverage. Additionally, clarification will be made for the use of MAR, as it was initially tested as a point source, however over the course of the study this evolved into a gridded use, and may not have been fully updated in writing as an oversight.

4. Some sections need to be rewritten to clear up ambiguity in what was done, what is being extrapolated, etc.

Line by line comments will be addressed to improve the manuscript and lessen any ambiguity in writing. Additionally, effort will be made to include much of the supplemental information in the main body of the manuscript so as to avoid further ambiguity with methods.

**Reviewer 2**

1. The abstract of the text needs to be re-written. When compared to the introduction it is of a lower standard of writing, and it doesn't convey the key findings well.

The abstract will be rewritten to cover findings more comprehensively, once all other comments are addressed.

2. When describing George VI Shelf (Lines 47-55), the authors need to do some wider reading of literature. For example, they should use past work here to describe the different glaciological settings of the north and south GVIIS. A study area figure is also required.

A study area figure will be produced and included, further efforts to discuss the differing settings of the north and south end of George VI ice shelf will be made. References such as Smith et al., (2007), Holt et al., (2013) and Hambrey et al., (2015) will be used among others to improve the quality of this discussion.

3. It makes little sense to me to use the NDWI Green and Near Infrared method over the NDWI blue and red method, given that the majority of literature would use the latter, and this has been well justified in many previous papers. I am not convinced as to why the authors chose to use this alternative thresholding method, and the text in S2 still does little to convince me. It would be interesting to see some maps showing the differences between the two thresholding approaches.

Inclusion of information from the supplement and associated MSc project will be added to this work. Comparisons between each NDWI methodology were made at the initiation of the study on a test region, where Green-NIR served to produce lake polygons which were more 'strict' towards lake shorelines, and therefore less likely to pick up slush and saturated snow surrounding lakes.

4. The lakes in some imagery were manually delineated, yet there is no mention of the error that should be considered when comparing these manually delineated lakes to lakes found using the thresholding method. Overall, the authors should consider the errors associated with all methods, and reference these where appropriate.

Consideration of error will be made for manual delineation, with the generated error being subject to discussion within the research group. This was tested again at the initiation of the project, however it was found that due to the subjectivity of manual delineation, a true value of error was hard to ascertain if multiple people carried out manual delineation, or the same person carried out the process over several days. Hence, further testing may be necessary.

5. The authors state that they use a different threshold for Landsat 1 because the bands do not correlate with the other Landsat instruments. But I question whether the Sentinel-2 bands correlate? If not, why did you not use a different threshold for that too?

Sentinel 2 and Landsat 8 bands correlate closely, and therefore the same threshold value was used for each satellite. This will be made clearer in the text. Landsat-1 was an anomaly as it includes many fewer bands of differing width to more modern instruments.

6. Is there full ice shelf coverage for every data point investigated? If not, how much of the ice shelf is 'missing'?

Full ice shelf coverage is found for all data points other than those specified. A diagram including this information was produced but not included in the text due to initial constraints on manuscript length. However this will be included with the changes made in response to comments.

7. The authors only show satellite imagery of GVIIS in maximum melt years, however they comment (Line 167) on the spatial organisation of surface meltwater in low melt years too. It would be useful to see some figures showing this, to allow the reader to see the changes that occur over time.

Included in the original MSc thesis was a series of diagrams showing the full lake coverage across GVIIS for each year during the study period. This will be included in the supplement in future submission as per this comment. However, it would not be appropriate as part of the main text due to the size of the diagram.

8. The authors suggest that they convert the areas for all data that wasn't affected by the Landsat-7 scan line failure, ultimately reducing the areas? This is a questionable decision as it broadly means the data presented is not representative of the true area of melt on GVIIS, which is an important statistic to have. I suggest the authors present both the converted and unconverted data.

We agree with the suggestion to approach Landsat-7 data in an alternative manner. Pre- and post- conversion data will be included in discussion in the updated manuscript. Another approach we have discussed would be to keep non-LS7 data as unaltered, and to convert Landsat-7 data using the 0.78 scaling factor. However, inclusion of both sets of values would show a more complete picture.

**References:**

Banwell, A. F. *et al.* (2021) 'The 32-year record-high surface melt in 2019/2020 on the northern George VI Ice Shelf, Antarctic Peninsula', *Cryosphere*, 15(2), pp. 909–925. doi: 10.5194/TC-15-909-2021.

Hambrey, M. J. *et al.* (2014) 'Structure and sedimentology of George VI Ice Shelf, Antarctic Peninsula: Implications for ice-sheet dynamics and landform development', *Journal of the Geological Society*, 172(5), pp. 599–613. doi: 10.1144/jgs2014-134.

Holt, T. O. *et al.* (2013) 'Speedup and fracturing of George VI Ice Shelf, Antarctic Peninsula', *Cryosphere*, 7(3), pp. 797–816. doi: 10.5194/tc-7-797-2013.

Moussavi, M. *et al.* (2020) 'Antarctic Supraglacial Lake Detection Using Landsat 8 and Sentinel-2 Imagery: Towards Continental Generation of Lake Volumes', *Remote Sensing 2020, Vol. 12, Page 134*, 12(1), p. 134. doi: 10.3390/RS12010134.

Smith, J. A. *et al.* (2007) 'George VI Ice Shelf: past history, present behaviour and potential mechanisms for future collapse', *Antarctic Science*, 19(1), pp. 131–142. doi: 10.1017/S0954102007000193.

---

## Author Comment (AC3)

_Responses to reviewers – Change in Supraglacial Lakes on George VI Ice Shelf, Antarctic Peninsula: 1973-2020_:

- **Reviewer 1**:
  - Point 1 - We tested supraglacial lake bathymetry and ice shelf topography using the TanDEM-X DEM product. We used this high resolution DEM product to investigate the form of lake basins in order to assess whether they formed in a typical lake basin shape, or the U shape proposed in comments from Reviewer 1. Through our analysis we find no convincing evidence that supraglacial lakes on the George VI ice shelf (GVIIS)form in U shaped depressions, but rather shallow graduated basins and V shaped depressions (in the case of drainage dolines) (response figure 1). In the light of this finding, it is clear that increased melting will increase the surface extent of supraglacial lakes in a somewhat consistent manner, rather than maintaining the same surface footprint until they overspill. This is a result of the lakes being predominantly present in shallow depressions on the ice shelf surface. Hambrey et al., (2015) discusses the surface topography of GVIIS, describing the features between which the lakes form as 'pressure ridges', which reach only up to 5 metres high. As such, we conclude that in the case of GVIIS, surface topography has only an impact on where water pools initially, but as melting continues, water quickly spreads across the surface of the ice shelf. Hence, we see no convincing evidence to lead us to consider surface topography more deeply or adjust the focus of this manuscript.

[Figure]

**Surface Elevation (m)**

(1)
(2)
(3)
(4)
(5)
(6)
(7)
(8)
(9)
(10)
(11)
(12)
(13)
(14)
(15)
(16)
(17)
(18)
(19)
(20)

**Distance along transect (m)**

*Figure 1: Diagram displaying lake bathymetry cross sections for an assortment of supraglacial lake features on George VI ice shelf in TanDEM-X high resolution product (Wessel, 2018). Transect lines are numbered and shown in red. Blue denotes the region in which water is found. All numeric values on charts are given as metres – along transect and surface elevation. Most of the features cover lake features, however feature 16 covers the core of a drainage doline. Background produced from Landsat-7 imagery taken 2013-01-14.*

- Point 2 - Added clarification in text with references determining reasoning behind using Green and NIR methodology. We made use of this method over Red and Blue methodology due to the red blue method excluding areas of water, particularly where lakes were conjoined in the interconnected network of George VI Ice Shelf. The methodology in this paper was based in principle on the work initially carried out by Stokes et al. (2019) on East Antarctica, and hence we aimed to use the same method in the study. Other works have used this methodology in a glaciological setting (for example, Watson et al., 2018), and more contemporary works have used a combination of both methods, as both have different regions in which they excel (Corr et al., 2021). Values in non-Landsat-7 imagery are no longer scaled to be comparable to Landsat-7.
- Point 3 We clarify that gridded MAR was used rather than a single point.
- Point 4 This comment has been addressed through line by line edits, and major edits from both reviewers. Many sections have had added clarification and the abstract has been re-written to more readily fulfil its purpose and give relevant information to the reader as a summary of the manuscript.
- Figure 2 (previously, figure 1) All comments are addressed accordingly. We removed labelled locations and included these in a study area map as a new "figure 1".
- Figure 3 (previously figure 2) Month information added to figures, imagery dates can be found in supplementary table 1. Legend has been altered but kept in the same place due to figure clutter being avoided.
- Figure 4 (previously, figure 3) The scatter plots in supplement are directly associated with this figure and discussed as such in the figure caption. However previously we attempted to include these scatter plots in this figure but the authors decided through discussion that they added no further information to the diagrams, and cluttered the image. Instead, we decided they should be included as additional information in supplement.
- Figure 5 (previously, figure 4) The lack of trend is addressed more clearly in text. We focus on a wider study region with these figures as GVIIS is not a closed system, and is impacted by surrounding climatic changes. The associated in-text section discusses the wider influence of Antarctic Peninsula climate on GVIIS, and thus we consider the wider scale maps to be relevant and appropriate for inclusion in the figure.
- **Reviewer 2**:
  - Point 1 - Abstract has been rewritten to bring up to standard, this was neglected over pre-submission edits.
  - Point 2 – A short section has been added to describe the glaciological setting of GVIIS, including references to literature.
  - Point 3 - Added clarification in text with references determining reasoning behind using Green and NIR methodology. We made use of this method over Red and Blue methodology due to the red blue method excluding areas of water, particularly where lakes were conjoined in the interconnected network of George VI Ice Shelf. The methodology in this paper was based in principle on the work initially carried out by Stokes et al. (2019) on East Antarctica, and hence we aimed to use the same method in the study. Other works have used this methodology in a glaciological setting (for example, Watson et al., 2018), and more contemporary works have used a combination of both methods, as both have different regions in which they excel (Corr et al., 2021).
  - Point 4 – We calculate error for supraglacial lake delineation in manual and automated methods together, due to the inherent connection of automated methods to manual. We find the error value to be ± 9%. Detail of this process and error has been explained in section 2.1, and numeric information added to Table 4.1 in the supplement. In addition, we have referred to the error value in all quoted values present in the manuscript.
  - Point 5 - We took the values used in Stokes et al., (2019) as a benchmark for thresholding, and worked from these. While Stokes et al., (2019) made use of the

same thresholding value for both Landsat 8 and Sentinel 2, we tested both sets of imagery separately for thresholding values. After testing they were found to fall on the same values, in line with findings from previous literature. A different thresholding value was used for Landsat 1 due to the large difference in available bands between this satellite and contemporary satellites such as Landsat 8 and Sentinel 2 (Ihlen & Zanter, 2019; USGS, 1979).

- o Point 6 – Addressed by a timeseries chart in supplement (Figure S1.2). No periods use incomplete imagery. Either complete imagery is acquired, or the period is skipped so as to avoid problematic variability in the results.
- o Point 7 – Diagrams of lake extent between 2000-2020 have been added to the supplement from the associated Master's thesis, allowing for visualisation of patterns of lake extent for those interested.
- o Point 8 – We have made edits to the approach to non-Landsat-7 data in regards to Landsat-7 scaling issues. All values now quoted in the text are true values, rather than those scaled by the 0.78 Landsat-7 scale factor. However, diagrams still represent scaled values to be brought in line with Landsat-7 for the sake of comparability. This information is also now specified in-text to address any misunderstanding.

References:

Corr, D., Leeson, A., McMillan, Zhang, C., Barnes, T.: An inventory of supraglacial lakes and channels across the West Antarctic Ice Sheet. Earth System Science Data, [preprint], https://doi.org/10.5194/essd-2021-257 , in review, 2021

Ihlen, V. and Zanter, K.: Landsat 7 (L7) Data Users Handbook, 2019.

Hambrey, M. J., Davies, B. J., Glasser, N. F., Holt, T. O., Smellie, J. L., and Carrivick, J. L.: Structure and sedimentology of George VI Ice Shelf, Antarctic Peninsula: implications for ice-sheet dynamics and landform development, J. Geol. Soc. London., 172, 599–613, https://doi.org/10.1144/JGS2014-134, 2015.

Stokes, C. R., Sanderson, J. E., Miles, B. W. J., Jamieson, S. S. R., and Leeson, A. A.: Widespread distribution of supraglacial lakes around the margin of the East Antarctic Ice Sheet, Sci. Reports 2019 91, 9, 1–14, https://doi.org/10.1038/s41598-019-50343-5, 2019.

USGS, Landsat Data Users Handbook, 1979

Watson, C. S., King, O., Miles, E. S., Quincey, D. J.: Optimising NDWI supraglacial pond classification on Himalayan debris covered glaciers, 217, 414-425, https://doi.org/10.1016/j.rse.2018.02.020, 2018.

Wessel, B., "TanDEM-X Ground Segment – DEM Products Specification Document", EOC, DLR, Oberpfaffenhofen, Germany, Public Document TD-GS-PS-0021, Issue 3.2, 2018 [Online]. Available: https://tandemx-science.dlr.de/ 2018